# Genome-Wide Analysis of the WRKY Gene Family in Malus domestica and the Role of MdWRKY70L in Response to Drought and Salt Stresses

**DOI:** 10.3390/genes13061068

**Published:** 2022-06-15

**Authors:** Ying Qin, Haixia Yu, Siyuan Cheng, Zhu Liu, Cheng Yu, Xiaoli Zhang, Xinjian Su, Jingwen Huang, Shuting Shi, Yangjun Zou, Fengwang Ma, Xiaoqing Gong

**Affiliations:** State Key Laboratory of Crop Stress Biology for Arid Areas/Shaanxi Key Laboratory of Apple, College of Horticulture, Northwest A&F University, Xianyang 712100, China; qinying883@163.com (Y.Q.); y17805659651@163.com (H.Y.); csybrandnew@163.com (S.C.); lz2020jy@163.com (Z.L.); yucheng99820@163.com (C.Y.); xlzhang0725@nwafu.edu.cn (X.Z.); 15837766774@163.com (X.S.); xn87095532@163.com (J.H.); shishuting@163.com (S.S.); zouyangjun@nwsuaf.edu.cn (Y.Z.); fwm64@nwsuaf.edu.cn (F.M.)

**Keywords:** apple, WRKY family, MdWRKY70L, drought stress, salt stress

## Abstract

The WRKY transcription factors are unique regulatory proteins in plants, which are important in the stress responses of plants. In this study, 113 WRKY genes were identified from the apple genome GDDH13 and a comprehensive analysis was performed, including chromosome mapping, and phylogenetic, motif and collinearity analysis. MdWRKYs are expressed in different tissues, such as seeds, flowers, stems and leaves. We analyzed seven WRKY proteins in different groups and found that all of them were localized in the nucleus. Among the 113 MdWRKYs, MdWRKY70L was induced by both drought and salt stresses. Overexpression of it in transgenic tobacco plants conferred enhanced stress tolerance to drought and salt. The malondialdehyde content and relative electrolyte leakage values were lower, while the chlorophyll content was higher in transgenic plants than in the wild-type under stressed conditions. In conclusion, this study identified the WRKY members in the apple genome GDDH13, and revealed the function of MdWRKY70L in the response to drought and salt stresses.

## 1. Introduction

The growth and development of plants are affected by various biotic and abiotic stresses. Abiotic stress is often caused by extreme environmental conditions, such as drought, low and high temperature, salt, soil nutrients and so on [1]. Plants adapt to, avoid, and overcome adverse environments through a variety of physiological and biochemical mechanisms. For example, when subjected to drought stress, plants change their rate of respiration and photosynthesis. They slow down transpiration by controlling stomatal conductance, thus retaining moisture [1,2,3]. Salt stress is one of the most important abiotic stressors, which leads to ion imbalance and water loss through osmotic reactions [4,5]. However, plants can cope with salt stress by synthesizing different osmotic substances, reducing the absorption of Na^+^, and some other ways [6,7].

WRKY constitutes one of the largest transcription factor families in plants, involved in various processes of growth, development and stress responses [8,9,10]. Although the DNA binding domain of WRKY proteins is highly conserved, the overall structure of the WRKY proteins varies widely and has been divided into different groups, reflecting their different biological functions. The WRKY domain consists of the conserved WRKYGQK amino acid sequence at its N-terminal and a novel zinc finger-like motif at its C-terminal. WRKY proteins with two WRKY domains belong to Group I, and proteins with one WRKY domain belong to Group II, which have been further divided into five subgroups based on the primary amino acid sequence. A small number of WRKY proteins have a single finger motif different from the members of Groups I and II. They contain a Cx7Cx23HxC motif rather than a Cx4–5Cx22–23HxH pattern in WRKY domain, and they are assigned to Group III [11,12,13].

Studies have shown that WRKY transcription factors are critical in regulating plant responses to pathogens, and a growing number of studies have reported that WRKYs are also involved in the regulation of abiotic stress responses in plants [8,14,15]. For example, overexpression of MdWRKY30 enhances salt stress tolerance in Arabidopsis thaliana and the apple callus [16]. TaWRKY genes improve the abiotic stress tolerance in transgenic A. thaliana, and there are four TaWRKYs which are comprehensive hubs of multiple stress signaling pathways in wheat [17]. Similarly, ectopic expression of FtWRKY46 enhances stress tolerance of transgenic plants by regulating the scavenge of reactive oxygen species (ROS) and the expression of stress-related genes [18]. Overexpression of IbWRKY2 and HbWRKY83 improved the tolerance of transgenic Arabidopsis plants to salt and drought stress by enhancing ROS elimination [19,20]. In tomato, WRKY8 plays a regulatory role in pathogen defense responses as well as in drought and salt stress responses [21]. The AhWRKY75 gene conferred salt stress tolerance in transgenic peanut plants by improving the efficiency of the ROS scavenging and photosynthesis under stress treatments [22]. Understanding the function and evolution of WRKY transcription factors will help to identify common connections in complex signaling pathways, to promote improvements in agricultural crop yield and quality [14].

Although the WRKY genes in apples have been investigated in many studies, the genetic information has been updated along with the release of apple genome GDDH13. Therefore, we re-identified the WRKY genes in the apple genome and carried out a comprehensive bioinformatics analysis of this family’s members. The expression pattern of MdWRKYs was analyzed according to the shared seq-data on-line. In addition, transgenic tobacco plants were generated to analyze the biological role of MdWRKY70L, induced by different stresses, under drought and salt stresses. Our results will be beneficial for revealing the role of MdWRKY70L in apple responses to stressed conditions.

## 2. Materials and Methods

### 2.1. Identification and Comprehensive Analysis of the WRKY Family Members in the Apple Genome

HMMER was used to identify the WRKY genes in apple genome GDDH13. The WRKY domain (PF03106) file in the HMM raw format was downloaded from the Pfam database and used as the default query sequence to search candidate WRKY sequences in the apple genome [23,24]. All of the obtained sequences were subjected to the CD search program (https://www.ncbi.nlm.nih.gov/Structure/cdd/wrpsb.cgi, accessed on 5 September 2021) and SMART (http://smart.embl-heidelberg.de/, accessed 5 September 2021) to verify their reliability as target WRKY genes. Sequences without the intact WRKY domain were removed from the candidates [25]. The physical and chemical properties of MdWRKYs, such as the molecular weight and isoelectric point, were analyzed by ExPASy (https://www.expasy.org/, accessed on 8 September 2021). Duplicate gene classifiers were analyzed using MCScanX. Chromosome localization was mapped using MapChart, and MEME (https://meme-suite.org/meme/index.html, accessed on 12 September 2021) was used for the motifs analysis. Multiple sequence alignments were carried out using COBALT (https://www.ncbi.nlm.nih.gov/tools/cobalt/cobalt.cgi, accessed on 15 September 2021) with the default values. The phylogenetic tree was constructed using the neighbor-joining (NJ) method and visualized with iTOL [26]. Cis-acting elements were predicted via PlantCARE (http://bioinformatics.psb.ugent.be/webtools/plantcare/html/, accessed on 5 October 2021), and the gene structure was analyzed via TBtools [27]. The homologous relationship map between the homologous WRKY genes of apple and Arabidopsis was constructed using Dual Synteny Plotter software [27], and the homologous map was also constructed with Dual Synteny Plotter software. The expression data of MdWRKYs were downloaded from Apple MDO (http://bioinformatics.cau.edu.cn/AppleMDO/index.php, accessed on 1 November, 2021), and exhibited in the form of a heatmap. All the results above were visualized using TBtools.

### 2.2. Plant Materials and Stress Treatments

Tissue-cultured ‘GL-3′ (Malus domestica) plants were used to analyze the expression of MdWRKY70L under different stresses. The plants were transferred to plastic pots and cultured in a greenhouse with regular watering for one month before the treatments began. The drought treatment consisted of withholding water, and leaves were sampled at 0, 2, 4, 6, 8, and 10 d. For the salt and abscisic acid (ABA) treatments, we added 200 mM NaCl or 100 mM ABA to the irrigation solutions and sampled leaves at 0, 2, 4, 6, 8, 12, and 24 h. To induce alkaline stress, Na_2_CO_3_/NaHCO_3_ (200 mM) was added to the irrigation solution, and leaves were sampled at 0, 1, 3, 6, 12, and 24 h. Another group of plants were transferred to a 4 °C incubator for 24 h to induce chilling stress, and leaves were sampled at 0, 2, 4, 6, 8, 12, and 24 h. All the treatments were performed with three biological replicates separately, with five plants in each replicate. The sampled leaves were immediately frozen in liquid nitrogen and stored at −80 °C.

Tobacco plants (Nicotiana benthamiana) were directly seeded in a substrate and grown in an incubator at 25 ± 2 °C under 16 h of light (55–75 μmol m^−2^ s^−1^) and 8 h of dark. They were used to analyze the subcellular localization of selected WRKYs 4 weeks after germination. Nicotiana nudicaulia plants were prepared and cultured as described as Gong et al. (2015) [28]. Briefly, after sterilizing with 70% alcohol and HCLO, seeds were sown on MS medium. Then, 5 weeks later, the seedlings were used in different experiments.

### 2.3. Subcellular Localization Analysis of MdWRKYs

To confirm the subcellular localization of the MdWRKYs, the coding sequence (CDS) of 7 MdWRKYs were cloned into the pClone007 intermediate vector and then cloned into the pCAMBIA2300-EGFP vector to obtain MdWRKYs-GFP fusion proteins. The pCAMBIA2300-EGFP empty vector was used as the negative control. All the recombined vectors were transformed into Agrobacterium tumefaciens GV3101 via the freeze–thaw method. The transformed GV3101 were then injected into the leaves of N. benthamiana. The nuclear stain DAPI was injected into the samples 5 min earlier and the localizations were observed by confocal microscopy. All the primers are shown in Appendix A.

### 2.4. Genetic Transformation of MdWRKY70L

The CDS of MdWRKY70L was constructed into the pGWB411 vector and introduced into GV3101 via the freeze–thaw method. Transformation of N. nudicaulia was performed using the leaf disc transformation method according to Horsh et al. (1985) [29]. After infection and screening, the resistant buds were used to extract DNA to identify the positive transformed buds. The expression level of MdWRKY70L in the transgenic lines were analyzed by PCR analysis, then 5# and 6# were selected for subsequent experiments.

### 2.5. Resistance Analysis of Transgenic Tobacco

Five-week-old potted tobacco seedlings were used for the salt and drought stress treatments. For inducing drought stress, we stopped watering tobacco plants for 10 days. The seedlings were irrigated with 200 mM NaCl solution to induce salt stress. Phenotypes before and after the stress treatments were photographed and preserved. Samples were collected to determine the physiological indicators. Chlorophyll and the relative electrolyte leakage (REL) were determined according to Dong et al. (2020) [30]. Malondialdehyde (MDA) levels were measured by the barbiturate method using an MDA test kit (Nanjing Jiancheng, Nanjing, China). The relative water loss was measured according to Gong et al. (2015) [28], as well as the DAB and NBT staining. The stress treatments were performed with at least three biological replicates with identical plant numbers in each replicate.

### 2.6. Quantitative PCR (qPCR)

Total RNA was extracted using the WoLact^®^ Plant Total RNA Isolation Kit (Wolact, Hong Kong, China). The cDNA was synthesized using a Reverse Transcription Kit (Thermo Scientific, Waltham, MA, USA). The expression pattern of MdWRKY70L under different treatments was analyzed by qPCR on a LightCycler 96 quantitative instrument (Roche, Switzerland). MdMDH was used as the internal reference to calculate the relative expression of MdWRKY70L using the ^ΔΔ^CT method. Four replicates were performed for each sample. The primers are shown in Appendix A.

### 2.7. Statistical Analysis

The experimental data were analyzed using SAS 8.1 software (SAS Institute, Cary, NC, USA). Significant differences were detected with Duncan’s test. Asterisks indicated that the value was significantly different between OE and WT lines (* *p* < 0.05, ** *p* < 0.01).

## 3. Results

### 3.1. Identification of WRKY Genes in Apple Genome

According to the conserved domain of the WRKY protein, 113 WRKY sequences were identified from the apple genome by HMMER. The basic information of these 113 MdWRKYs is shown in Table 1. The CDS length of MdWRKYs ranged from 243 bp (MD12G1129000) to 2988 bp (MD07G1261100), encoding proteins from 80 aa to 995 aa. The molecular weights of the 113 predicted proteins were between 9.282 kDa (MD12G1129000) and 113.187 kDa (MD07G1261100). The isoelectric point values were between 4.81 (MD00G1140800) and 9.99 (MD13G1239100). Most WRKY proteins were located in the nucleus, and a few were located in the cell membrane or outside the cell. The duplication mode of most of the WRKYs were WGD/segmental, and only a few were tandem, proximal, or dispersed.

### 3.2. Chromosome Distribution and Evolutionary Analysis of the WRKY Sequences

We downloaded the distribution information of all the 113 MdWRKY sequences from the apple genome database, and displayed their physical sites on different chromosomes using MapChart software. The results showed that 110 out of the 113 sequences were distributed on the 17 chromosomes of apple, while the other three failed to anchor on any chromosome, including MD00G1140800, MD00G1143500 and MD00G1143600. Chromosome 7 had the most numbers of WRKYs, there were ten of them. Chromosome 2 had the least numbers of WRKYs, only two (Figure 1).

We employed 71 AtWRKYs of Arabidopsis and all the MdWRKYs to construct a phylogenetic tree. As shown in Figure 2, the 184 WRKY proteins were divided into three groups and seven subgroups. The MdWRKY proteins cluster into the same group shared similar conserved domain (Appendix A). Significant differences were observed between Arabidopsis and M. domestica. Most of the terminal branches of the phylogenetic tree connected two AtWRKYs or two MdWRKYs, except for four couples, including AtWRKY1 and MD12G1129000 in Group I, AtWRKY12 and MD07G1110400 in Group IIc, AtWRKY51 and MD15G1331300 in Group IIc, AtWRKY15 and MD02G1177500 in Group IId.

To further analyze the gene replication relationship of WRKYs, comparative genomics analysis was carried out. The evolutionary relationships between Arabidopsis and apple were analyzed (Figure 3). The results showed that 31 out of 113 WRKYs of apple had 39 pairs of WRKY collinearity with Arabidopsis (Appendix A). Most of them had just one pair, while there were two pairs of MdWRKY70L collinearity between apple and Arabidopsis. The results showed that the WRKY gene family members of different species may come from the same ancestor and play a similar role in plants. It is noteworthy that there was no collinearity relationship between MdWRKYs located on chromosome 14 and 16 and AtWRKYs, suggesting that these WRKYs in the apple genome may be unique regarding its evolution.

### 3.3. Structural Analyses of the WRKY Sequences

The biological functions of proteins are often related to their unique structures, such as protein motifs and domains, and cis-elements on the promoters. To better understand the diversity of MdWRKYs, we analyzed their structural differences on DNA and protein levels. As shown in Figure 4, six conserved motifs were found in MdWRKYs (Appendix A). MdWRKYs in the same group had similar protein motifs, while differences were detected among different groups. For example, all the six motifs could be detected in Group I, while only motif 1, 2, and 3 could be detected in Group III (Figure 4B). There were different numbers of introns distributed on genomic DNA of MdWRKYs, a few members had only one intron and most of them had two to four introns (Figure 4C).

### 3.4. Expression Profile of MdWRKY Genes

Most WRKY members in Figure 5 expressed highly in the detected tissues, including seed, flower, stem and leaf. For example, MD15G1106600, MD03G1057400, MD00G1140800 and MD08G1127200 expressed highly in the four tissues. There were also some members whose expression levels were obviously lower than others in the four tissues, for example, MD01G1210200, MD15G1419600, and Md12G1243400. Among all the detected MdWRKYs, MD08G1127200 was highly expressed in the seed. The expression of MD03G1057400 was highest in the flower and the leaf. The expression of MD15G1106600 was highest in the stem. MdWRKY70L (MD01G1168600) also expressed highly in the four tissues, and the highest expression was observed in the stem and the lowest expression was observed in the seed, implying that it may also be involved in plant growth and development (Figure 5).

We analyzed the cis-elements on the promoters of MdWRKYs and found that most of them can be induced by stressed conditions, as we observed cis-elements related to drought, low temperature and salt stresses (Figure 4D). We previously analyzed expression levels of some MdWRKYs under different stresses, and found a member from Group III, MD01G1168600, was induced by drought, salt, alkali, low temperature stresses, as well as ABA treatment, suggesting that it might be involved in the responses to these stressed conditions in apples (Figure 6). MD01G1168600 was clustered closely with AtWRKY54 and AtWRKY70 (Figure 2), so we named it MdWRKY70L. 

### 3.5. Subcellular Localization of MdWRKYs Proteins

To investigate the subcellular localization of MdWRKYs, several members were recombined with GFP protein to construct MdWRKYs-GFP fusion proteins, and transiently expressed in the leaves of tobacco (N. benthamiana). The GFP fluorescence in tobacco leaves was observed using a confocal laser scanning microscope, and all the fusion proteins were observed in the nuclei, while the GFP protein was detected both in the nuclei and in the cytoplasm (Figure 7), suggesting that the seven MdWRKYs we selected were localized in the cell nucleus, including MdWRKY70L.

### 3.6. MdWRKY70L Enhanced Drought and Salt Stress Tolerance in Transgenic Tobacco Plants

Previously, we found that MdWRKY70L was induced by drought, salt and ABA treatments. To further analyze its biological role in plants under stresses, we generated transgenic tobacco plant overexpressed MdWRKY70L. We obtained several positive transgenic lines, and 5# and 6#, with different expression levels of MdWRKY70L, were used in the stress treatments (Appendix A). When the seedlings were 5-weeks old, they were exposed to drought stress treatment via controlling the water. When watering was stopped for 10 days, the leaves of WT tobacco plants wilted and turned yellow, and their growth was inhibited, while 5# and 6# plants were less affected by the same treatment (Figure 8A). Under drought stress, the chlorophyll content was higher, while the REL, MDA content and relative water loss were lower in OE plants than in WT plants, indicating that the damage experienced by OE plants was much slighter than that experienced by WT plants (Figure 8C–F). In addition, leaves of WT plants were stained deep brown and blue with DAB and NBT, respectively, while the colors were lighter in leaves of OE plants, suggesting that OE plants accumulated less ROS under drought stress (Figure 8B).

We also treated WT and OE plants with salt stress, and obvious yellowing was observed on WT leaves, while leaves of OE plants were still green (Figure 8G). Higher chlorophyll content was detected in OE plants than in WT plants (Figure 8I). The REL was also higher in WT than in OE plants after salt stress treatment (Figure 8J). The DAB and NBT staining of the leaves of WT plants were stronger than was observed in the leaves of OE plants (Figure 8H), indicating that the WT plants were much more seriously damaged than the OE plants under salt stress. In conclusion, all these results suggested that overexpression of MdWRKY70L effectively improved stress tolerance of transgenic tobacco to drought and salt stresses.

## 4. Discussion

WRKY transcription factors are key regulators in many biological processes in plants. Great progress has been made in plants to reveal their function, for example, many WRKY genes have been proved to improve plant stress tolerance [15,31]. With the development of molecular techniques, members of this gene family have been discovered in an increasing number of varied species. 74 WRKYs have been identified in Arabidopsis, 109 in rice [32,33], 104 in poplar [34], 102 in flax [35], 45 in Eucommia ulmoides [36], 57 in melon [37], 86 in barley [38], and 64 in Isatis indigotica [39]. Previous study also revealed 127 WRKY members in apple [25]. Here, we identified 113 WRKY members using a local HMMER search and NCBI CDD verification, which were divided into three groups, 20 members in Group I, 95 in Group II, which were further divided into five subgroups, and 18 in Group III (Table 1). WRKY proteins were highly conserved in their amino acid sequences (Appendix A), and those in the same group had the same protein motifs. For example, motif 1 was found in all MdWRKYs, while motif 4 was only found in Group I members (Figure 4). The WRKY DNA binding domain was based on the WRKYGQK heptapeptide with one or two different amino acid(s), and that is exactly the core sequence of motif 1 (Appendix A). WRKYGKK and WRKYGMK, which we found in Group IIC members, were also the common optional form of WRKY binding domain, which has been reported by other studies [15,31,40] (Appendix A). In addition, some WRKY proteins contained a glutamate enrichment domain, some a proline enrichment domain, and others a leucine zipper structure [15,41,42]. These various domains enable WRKY proteins to play different roles in regulating gene expression [41]. In addition, the reasonable grouping and classification were also conducive to analyzing the regulatory function of WRKY transcription factors.

We analyzed the expression patterns of many WRKY members according to on-line data, and found differences in their expression in different tissues (Figure 5). They also had different stress-related cis-elements in their promoters (Figure 4), indicating that MdWRKYs might also participate in various stress responses as WRKY proteins in other plant species [15,43]. Here we found that the expression of MdWRKY70L changed as treatment time extended during the drought, salt, ABA, alkali, and low-temperature treatments (Figure 6). Among them, the expression of MdWRKY70L was up-regulated in a short time after drought and salt stressors were initiated, suggesting that MdWRKY70L might be an important regulator in drought and salt stress responses in apples. Many studies have reported the involvement of WRKYs in stresses. For example, EjWRKY17 enhances drought resistance in transgenic A. thaliana, and TaWRKY46 enhances osmotic stress tolerance in transgenic A. thaliana [43,44]. Transcription factors are usually located in the nucleus where they perform transcriptional regulatory functions [45]. Although some WRKY members have been predicted to be located outside the nucleus, we randomly selected seven WRKYs to detect their localization and found that they were all located in the nucleus (Figure 7). The predicted location in membranes or organelles may be due to their interactions with other proteins or some stimuli, resulting in a change in location [46,47]. For example, the transcription factor BZR1 is located in the cytoplasm in the absence of any treatment but it is recruited into the nucleus after brassinolide treatment [46].

MdWRKY70L was found collinear with AtWRKY70 (AT3G56400) and AtWRKY54 (AT2G40750) (Figure 4). They tended to be highly similar in structure and function [48]. Research has shown that the closer the clustering relationship is in a phylogenetic tree, the more likely the members are to have similar functions [49,50]. It has been reported that FtWRKY46 and GhWRKY41 are homologous genes, both of them enhancing salt stress tolerance in transgenic plants [18,51]. AhWRKY75 is closely related to AtWRKY75 and both improve salt stress tolerance in plants [22]. In this study, of MdWRKY70L and AtWRKY54, 70 were clustered in the same branch (Figure 2), suggesting that MdWRKY70L may have similar functions as AtWRKY54 or AtWRKY70. Studies have shown that WRKY54 is involved in the drought stress response in A. thaliana [52]. The homologous gene of AtWRKY70 in citrus, FcWRKY70, functions in transgenic plants responding to drought stress, too [28]. The expression of MdWRKY70L was also up-regulated by drought stress, it might also be involved in the drought stress response as AtWRKY54 and FcWRKY70. Thus, we generated transgenic tobacco plant overexpressed MdWRKY70L and showed that it indeed improved drought stress tolerance, as well as salt stress tolerance in transgenic tobacco plants (Figure 8).

Under stressed conditions, REL and ROS reflect damage in plants to a certain extent [53,54]. We found that NBT and DAB staining were weaker in OE plants than in the WT under both drought and salt stress treatments (Figure 8B,H). MDA is the final decomposition product of membrane lipid peroxidation, and its content reflects the degree of plant stress. The membrane lipid peroxidation level of drought-tolerant plants is lower than that of non-drought-tolerant plants [29,55]. Our results showed that the MDA content in OE plants was lower than that in the WT plants after the drought treatment (Figure 8E). Taken together, we demonstrated that MdWRKY70L improved the drought and salt stress tolerance in plants. Exogenous ABA improves salt stress tolerance in Lonicera lonicera and drought stress tolerance in Gynura cusimbua [56,57]. Studies have demonstrated that ABA-induced WRKY gene expression is often related to drought and salt stress [15,58,59]. AtWRKY33 contains an ABRE element, its induction is dependent on ABA signaling. Overexpression of AtWRKY33 increases salt stress tolerance in Arabidopsis [59]. FcWRKY40 is up-regulated by ABA and salt, and confirmed as a target of FcABF2, an ABA response element binding factor 2, in Fortunella crassifolia. Overexpression of FcWRKY40 functions positively in salt stress tolerance in transgenic plants [60]. MdWRKY70L was also induced by ABA treatment and contained ABA response elements in its promoter (Figure 4 and Figure 6). Therefore, we hypothesized that MdWRKY70L might function in transgenic plants responding to drought and salt stress in the same way. However, many studies have shown that WRKY transcription factors respond to stressed conditions in a variety of ways, so the specific mechanism of MdWRKY70L in regulating drought and salt stress responses remains to be further studied.

## 5. Conclusions

In summary, 113 members of WRKY transcription factors were identified in the apple genome. They were clustered into three groups and seven subgroups with different conserved protein motifs and gene structures. Most MdWRKYs were close to another WRKY member from the apple plant, except for MD12G1129000, MD07G1110400, MD15G1331300 and MD02G1177500. They were closer to AtWRKYs in the phylogenetic tree. The synteny analysis showed that MdWRKYs are located on chromosome 14 and 16 in the apple genome and this may be unique to apple evolution. In addition, a member in Group III, MdWRKY70L, was screened in response to multiple stresses. It was located in the nucleus and expressed in the stem at high level and in the seed at low level. Meanwhile, MdWRKY70L was collinear with At3G56400 and At2G40750 in A. thaliana, indicating this gene was conserved in plants. The phenotypic and physiological profiles demonstrated that overexpression of MdWRKY70L enhanced the resistance of transgenic tobacco plants to drought and salt stresses. In conclusion, this study comprehensively analyzed WRKY members in the apple genome, and revealed the function of MdWRKY70L in response to drought and salt stresses, which provided insight into the function of WRKY transcription factors in apples under stressed conditions.

## Figures and Tables

**Figure 1 genes-13-01068-f001:**
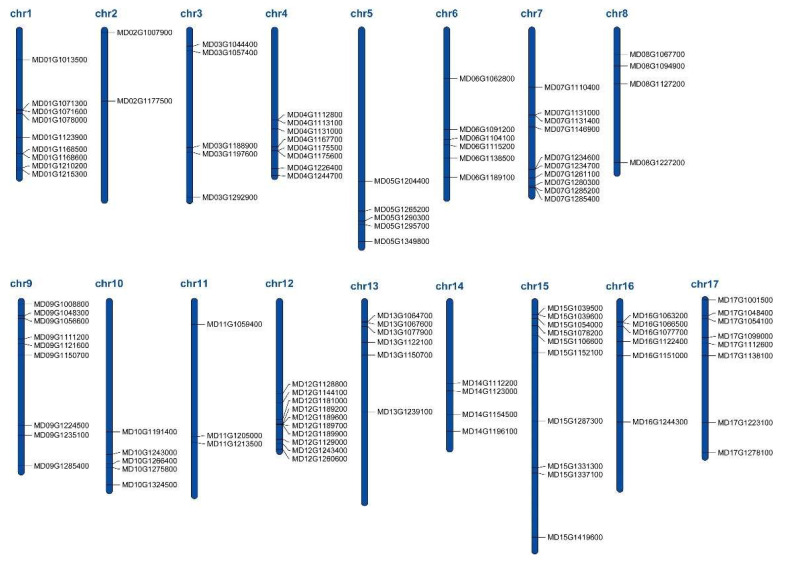
The distribution of WRKY genes on apple chromosomes. Chromosome number is indicated at the top of each chromosome.

**Figure 2 genes-13-01068-f002:**
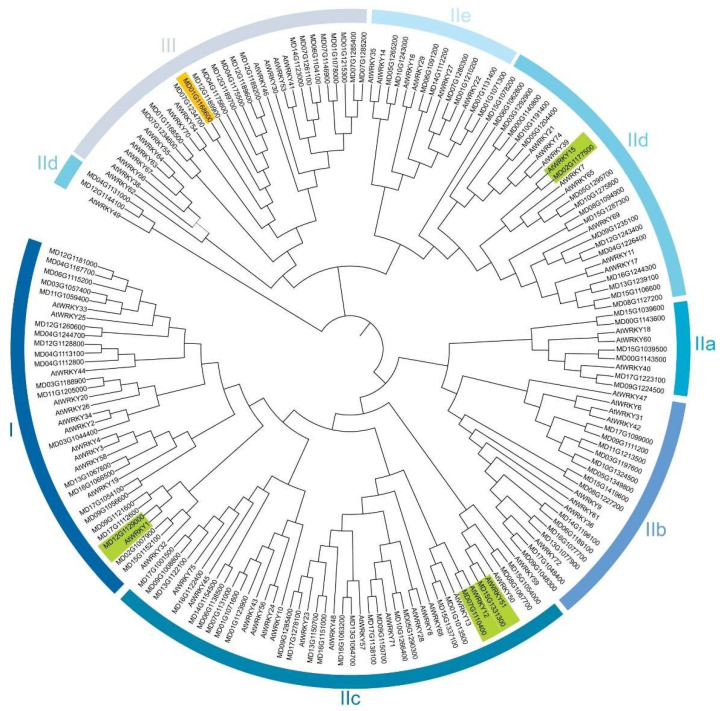
Phylogenetic tree of identified MdWRKY and AtWRKY proteins. Phylogenetic tree was constructed using MEGA-X program with the neighbor-joining (NJ) method. WRKY genes in each subgroup were shown with different colored arc. There were 71 AtWRKYs used in the figure and the sequence was downloaded from TAIR. The terminal branches of the phylogenetic tree connected an AtWRKY and an MdWRKY were marked with green. MdWRKY70L (MD01G1168600) was marked with orange.

**Figure 3 genes-13-01068-f003:**
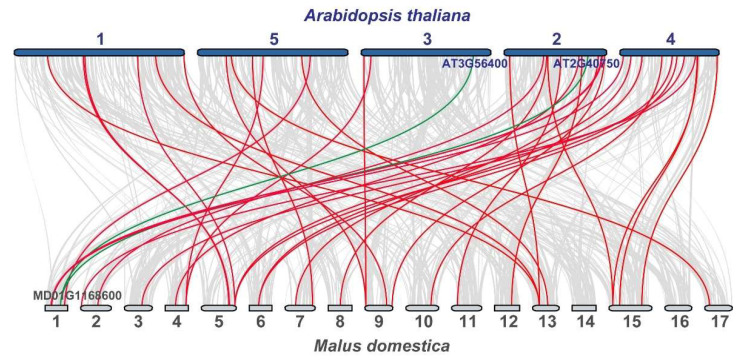
Synteny analysis of WRKY genes between Arabidopsis and Malus domestica. Syntenic WRKY genes of Arabidopsis (A. thaliana) and apple (M. domestica) are exhibited with red lines. Gray lines indicate the synteny blocks. The green lines indicate the synteny of MdWRKY70L (MD01G1168600) and AtWRKYs (AT3G56400 and AT2G40750).

**Figure 4 genes-13-01068-f004:**
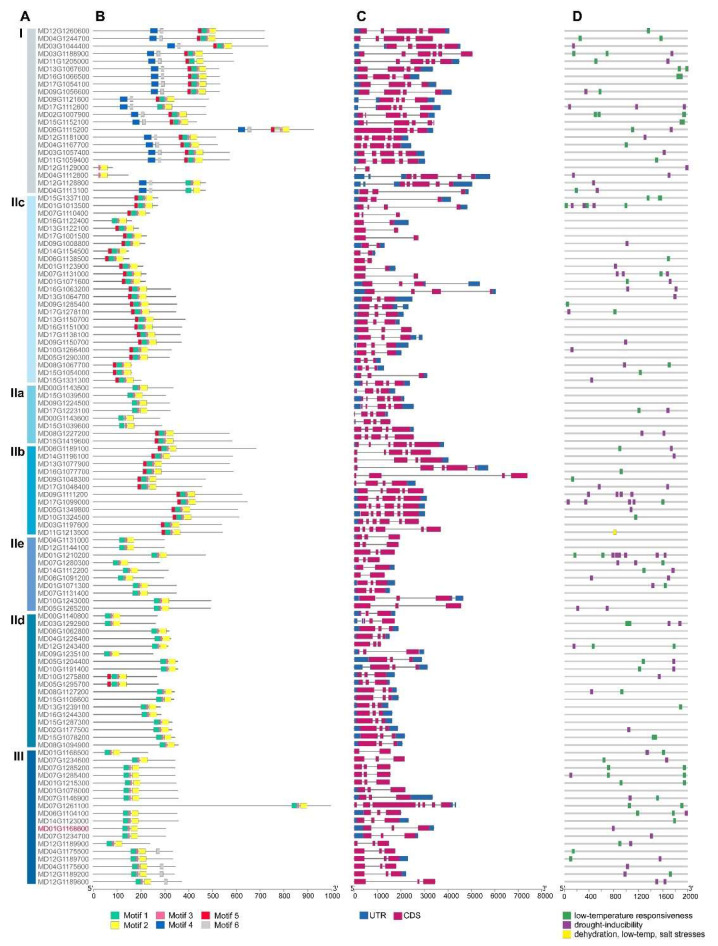
Phylogenetic tree, protein motif, gene structure and cis-acting element analysis of MdWRKY genes. (**A**) Phylogenetic tree. Phylogenetic tree was constructed using MEGA-X program with the neighbor-joining (NJ) method; (**B**) Protein motifs. The conserved motifs were identified by MEME. Each motif was represented by a colored box, the sequence information for each motif is provided in Appendix A; (**C**) Gene structures. Data in GFF3 format was downloaded from JGI and analyzed using TBtools. CDS, UTR and intron are represented by pink mauve box, dark blue box and grey line, respectively; (**D**) Cis-acting element prediction. Cis-element prediction was performed using PlantCARE. Only those related to stresses were selected for visualization and the rest were not displayed. The visualization of the results was achieved using TBtools. MdWRKY70L (MD01G1168600) are marked with red.

**Figure 5 genes-13-01068-f005:**
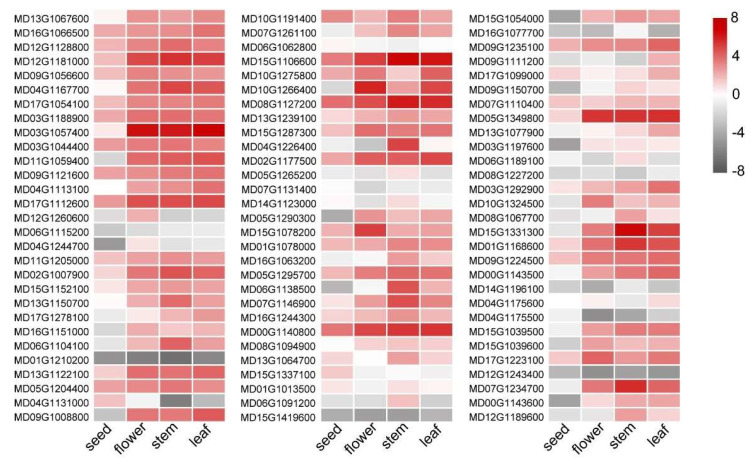
Expression analysis of MdWRKY genes in various tissues of apple. Expression profile data were obtained from Apple MDO database. The visualization of the result was achieved using TBtools.

**Figure 6 genes-13-01068-f006:**
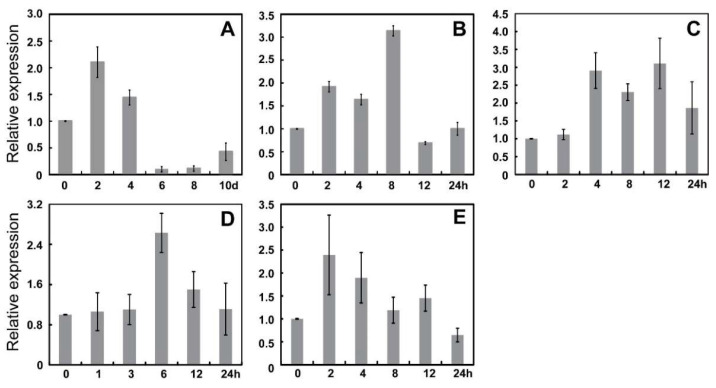
The expression pattern of MdWRKY70L under different stress treatments. (**A**) Drought stress treatment; (**B**) ABA treatment; (**C**) Salt stress treatment; (**D**) Alkali stress treatment; (**E**) Low temperature treatment. MdMDH was used as the internal reference and the expression data on 0 h or 0 d were normalized to “1”. The error bars indicate the standard deviation between four replicates.

**Figure 7 genes-13-01068-f007:**
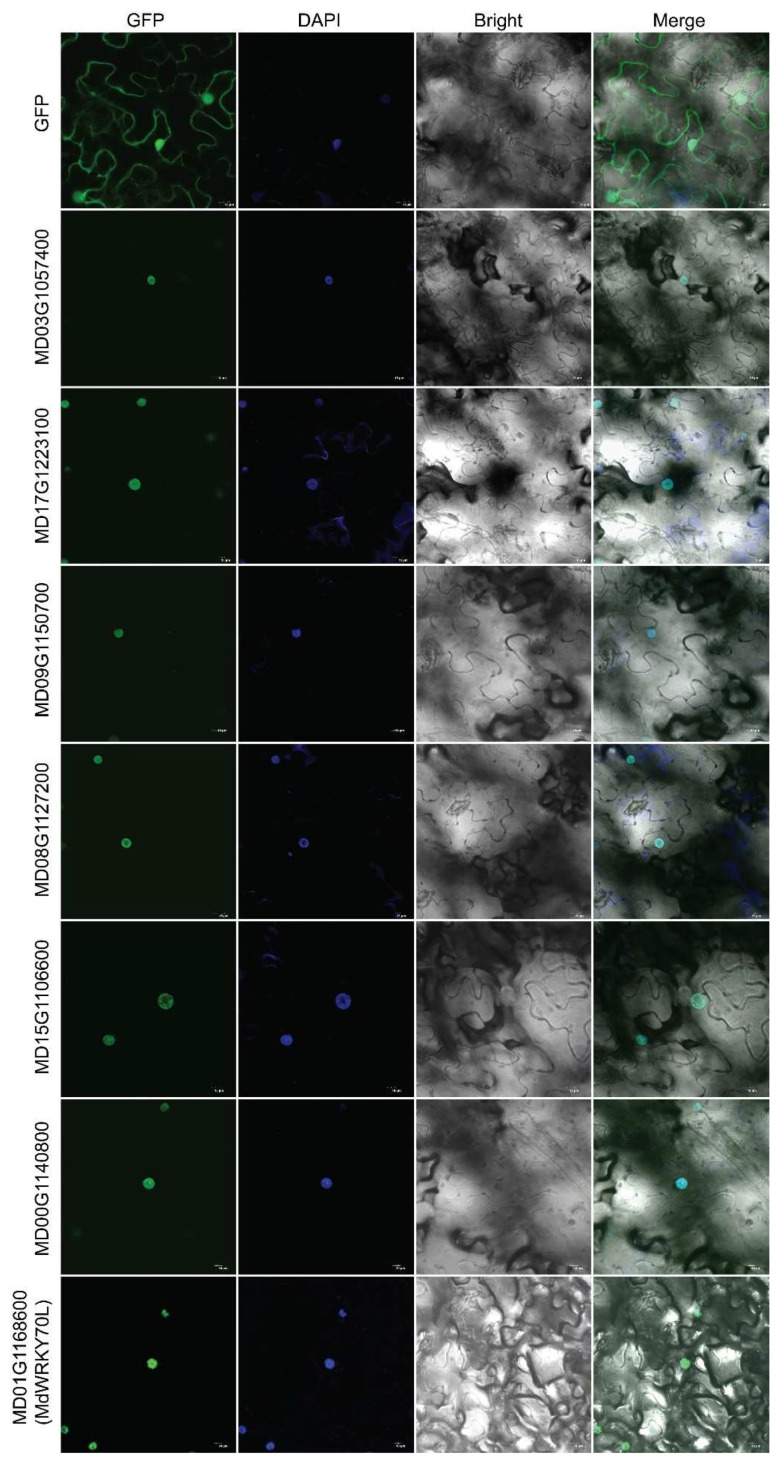
Subcellular localization analysis of seven MdWRKYs. The MdWRKY-GFP proteins and GFP proteins alone were expressed in tobacco leaf cells, respectively. The fluorescence was detected using a confocal microscopy under bright field and fluorescence. DAPI staining of cells was viewed under fluorescence, too. Scale bar = 15 µm.

**Figure 8 genes-13-01068-f008:**
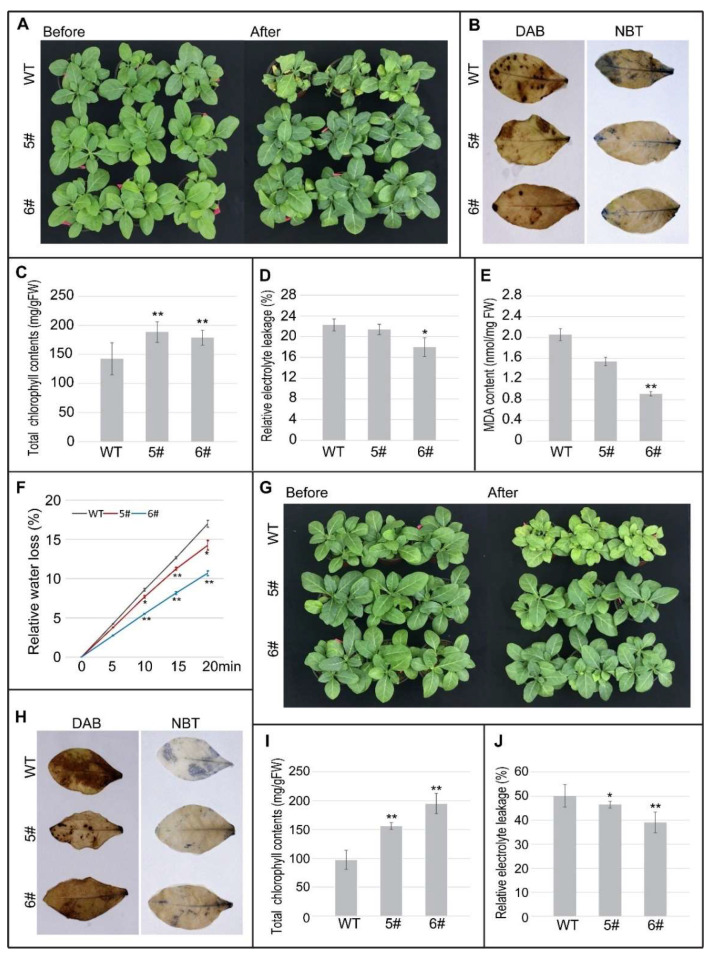
Overexpression of MdWRKY70L conferred enhanced drought and salt stresses tolerance in transgenic tobacco plants. (**A**) Phenotypes of 5-week-old seedlings withholding water for 10 days; (**B**) DAB and NBT staining after drought stress; (**C**) Chlorophyl contents; (**D**) The REL; (**E**) The MDA content; (**F**) Relative water loss; (**G**) Phenotype of 5-week-old seedlings after salt stress; (**H**) DAB and NBT staining after salt stress; (**I**) The chlorophyll content; (**J**) The REL. Asterisks represent significant difference between OE and WT lines (* *p* < 0.05, ** *p* < 0.01).

**Table 1 genes-13-01068-t001:** List of the identified WRKY genes in M. domestica with their detailed information.

Group	Genome No.	Chromosome No.	CDS(bp)	Amino Acid	MW (kDa)	Ip	Subcellular localization	Duplications
I	MD03G1057400	Chr03:4579079.4582074	1716	571	62.5	7.02	Nuclear	WGD/segmental
	MD11G1059400	Chr11:5068503.5071526	1719	572	62.9	6.77	Nuclear	WGD/segmental
	MD12G1181000	Chr12:26084482.26086791	1539	512	56.7	6.73	Nuclear	WGD/segmental
	MD04G1167700	Chr04:25792146.25794577	1563	520	57.7	7.10	Nuclear	WGD/segmental
	MD06G1115200	Chr06:25425907.25429273	2775	924	102.8	5.33	Nuclear	proximal
	MD12G1260600	Chr12:32647306.32651367	2157	718	79.0	6.40	Nuclear	WGD/segmental
	MD04G1244700	Chr04:32067852.32071212	2154	717	78.5	5.99	Nuclear	WGD/segmental
	MD12G1128800	Chr12:20397217.20403016	1413	470	51.5	9.00	Nuclear	WGD/segmental
	MD04G1113100	Chr04:19846116.19851149	1413	470	51.5	8.93	Nuclear	proximal
	MD03G1188900	Chr03:25924164.25929514	1755	584	63.9	5.97	Nuclear	WGD/segmental
	MD11G1205000	Chr11:29914448.29918933	1767	588	64.0	5.86	Nuclear	WGD/segmental
	MD03G1044400	Chr03:3511777.3516305	2199	732	79.5	5.96	Nuclear	dispersed
	MD13G1067600	Chr13:4637048.4640408	1581	526	57.3	7.26	Nuclear	WGD/segmental
	MD16G1066500	Chr16:4642014.4644789	1587	528	57.2	8.38	Nuclear	WGD/segmental
	MD17G1054100	Chr17:4198534.4202033	1593	530	57.4	8.39	Nuclear	WGD/segmental
	MD09G1056600	Chr09:3741354.3745516	1587	528	57.6	8.62	Nuclear	WGD/segmental
	MD09G1121600	Chr09:9379281.9382725	1455	484	53.2	5.93	Nuclear	WGD/segmental
	MD17G1112600	Chr17:9643152.9646837	1416	471	51.7	6.64	Nuclear	WGD/segmental
	MD02G1007900	Chr02:496443.499890	1422	473	52.0	8.82	Nuclear	WGD/segmental
	MD15G1152100	Chr15:11267047.11270466	1305	434	48.1	7.27	Nuclear	WGD/segmental
IIa	MD15G1039600	Chr15:2798151.2799702	861	286	32.1	8.15	Nuclear	tandem
	MD00G1143600	Chr00:31240990.31242443	837	278	31.3	8.85	Nuclear	tandem
	MD15G1039500	Chr15:2783151.2784898	909	302	33.6	7.10	Nuclear	WGD/segmental
	MD00G1143500	Chr00:31227045.31229422	1005	334	37.0	7.00	Nuclear	tandem
	MD17G1223100	Chr17:27209201.27211741	966	321	35.7	8.20	Nuclear	WGD/segmental
	MD09G1224500	Chr09:27404228.27406362	963	320	35.3	7.59	Nuclear	WGD/segmental
	MD04G1112800	Chr04:19827951.19828792	441	146	16.7	9.60	Extracellular	WGD/segmental
	MD12G1129000	Chr12:30451746.30454869	243	80	9.3	9.85	Cytoplasmic/ Nuclear/ Mitochondrial	proximal
IIb	MD17G1099000	Chr17:8402127.8405223	1938	645	70.0	6.42	Nuclear	WGD/segmental
	MD09G1111200	Chr09:8267211.8270166	1878	625	68.2	6.33	Nuclear	WGD/segmental
	MD05G1349800	Chr05:46759691.46762704	1821	606	65.3	7.67	Nuclear	WGD/segmental
	MD11G1213500	Chr11:31232215.31235905	1626	541	59.0	6.45	Nuclear	WGD/segmental
	MD03G1197600	Chr03:27003520.27006281	1617	538	59.0	6.42	Nuclear	WGD/segmental
	MD15G1419600	Chr15:52086817.52089361	1749	582	64.2	5.10	Nuclear	WGD/segmental
	MD08G1227200	Chr08:29356105..29358646	1713	570	62.312	5.35	Nuclear	WGD/segmental
	MD14G1196100	Chr14:28655099.28658370	1755	584	63.3	6.48	Nuclear	WGD/segmental
	MD06G1189100	Chr06:32608598.32612425	2052	683	73.5	7.23	Nuclear	WGD/segmental
	MD16G1077700	Chr16:5438696.5444408	1764	587	64.1	6.43	Nuclear	WGD/segmental
	MD13G1077900	Chr13:5478407.5482438	1716	571	61.9	5.95	Nuclear	WGD/segmental
	MD17G1048400	Chr17:3528939.3531555	1368	455	50.0	7.59	Nuclear	WGD/segmental
	MD09G1048300	Chr09:3166930.3174316	1413	470	51.2	7.61	Nuclear	WGD/segmental
	MD10G1324500	Chr10:40512430.40515449	1836	611	65.7	6.61	Nuclear	WGD/segmental
IIc	MD15G1054000	Chr15:3683177.3684465	486	161	18.2	5.65	Nuclear	WGD/segmental
	MD08G1067700	Chr08:5387244.5388373	486	161	18.2	8.53	Nuclear	WGD/segmental
	MD15G1331300	Chr15:36715532.36718649	600	199	22.3	5.86	Nuclear	dispersed
	MD07G1110400	Chr07:12683567.12688395	711	236	26.8	8.17	Nuclear	dispersed
	MD01G1013500	Chr01:6515683.6519808	813	270	30.4	8.93	Nuclear	WGD/segmental
	MD15G1337100	Chr15:37891029.37895909	816	271	30.2	8.93	Nuclear	WGD/segmental
	MD05G1290300	Chr05:42212192.42214208	960	319	35.1	6.60	Nuclear	WGD/segmental
	MD10G1266400	Chr10:35935449.35937762	984	327	36.0	6.46	Nuclear	WGD/segmental
	MD09G1150700	Chr09:11892767.11895678	1110	369	41.4	7.75	Nuclear	WGD/segmental
	MD17G1138100	Chr17:12392128..12394598	1098	365	41.2	6.76	Nuclear	WGD/segmental
	MD13G1064700	Chr13:4465565.4471604	1041	346	37.7	6.59	Nuclear	WGD/segmental
	MD16G1063200	Chr16:4485285.4490649	975	324	35.5	6.60	Nuclear	WGD/segmental
	MD16G1151000	Chr16:11906129.11908067	1116	371	41.1	5.60	Nuclear	WGD/segmental
	MD13G1150700	Chr13:11807823.11809934	1158	385	42.7	6.11	Nuclear	WGD/segmental
	MD17G1278100	Chr17:33812003.33814316	1041	346	38.2	6.27	Nuclear	WGD/segmental
	MD09G1285400	Chr09:36364455.36366937	1053	350	38.4	5.85	Nuclear	WGD/segmental
	MD01G1123900	Chr01:23690319.23691083	627	208	23.6	8.91	Nuclear	dispersed
	MD01G1071600	Chr01:17669166.17671888	657	218	24.6	9.36	Nuclear	WGD/segmental
	MD07G1131000	Chr07:18748505..18750267	669	222	24.9	9.36	Nuclear	WGD/segmental
	MD06G1138500	Chr06:28356819.28357724	453	150	17.1	9.56	Nuclear	WGD/segmental
	MD14G1154500	Chr14:24914608.24915904	447	148	17.1	9.59	Extracellular	WGD/segmental
	MD16G1122400	Chr16:8821613.8823564	486	161	18.0	9.08	Nuclear	WGD/segmental
	MD13G1122100	Chr13:8993050.8995368	573	190	21.7	9.74	Nuclear	WGD/segmental
	MD09G1008800	Chr09:616602.619344	651	216	24.8	9.14	Nuclear	WGD/segmental
	MD17G1001500	Chr17:91553.93431	672	223	25.6	9.14	Nuclear	WGD/segmental
IId	MD15G1078200	Chr15:5334685.5336858	1029	342	37.2	9.26	Nuclear	WGD/segmental
	MD06G1062800	Chr06:10770799.10772691	957	318	35.9	9.58	Nuclear	dispersed
	MD10G1191400	Chr10:28819078.28822211	1068	355	40.0	9.68	Nuclear	WGD/segmental
	MD05G1204400	Chr05:33402565.33405458	1065	354	40.0	9.70	Nuclear	WGD/segmental
	MD02G1177500	Chr02:15663260.15665127	993	330	36.0	9.65	Nuclear	WGD/segmental
	MD08G1094900	Chr08:7968389.7970451	1071	356	38.7	9.41	Nuclear	WGD/segmental
	MD15G1287300	Chr15:26365198.26366819	996	331	36.2	9.54	Nuclear	WGD/segmental
	MD12G1243400	Chr12:31414877.31416020	942	313	35.3	9.96	Nuclear	WGD/segmental
	MD04G1226400	Chr04:30603205.30604718	978	325	36.6	9.77	Nuclear	WGD/segmental
	MD15G1106600	Chr15:7467131.7469028	1017	338	36.7	9.46	Nuclear	WGD/segmental
	MD08G1127200	Chr08:11928202.11930014	1026	341	37.0	9.32	Nuclear	WGD/segmental
	MD16G1244300	Chr16:26579453.26581081	855	284	30.9	9.87	Nuclear	WGD/segmental
	MD13G1239100	Chr13:24320203.24321656	846	281	30.7	9.99	Nuclear	WGD/segmental
IIe	MD04G1131000	Chr04:21800375.21802338	894	297	33.4	5.72	Nuclear	WGD/segmental
	MD12G1144100	Chr12:22324583.22326472	894	297	33.1	5.94	Nuclear	WGD/segmental
	MD03G1292900	Chr03:36975267.36977012	783	260	30.3	5.17	Nuclear	WGD/segmental
	MD00G1140800	Chr00:30743431.30745181	807	268	30.8	4.81	Nuclear	WGD/segmental
	MD05G1295700	Chr05:43023949.43025466	822	273	30.2	5.80	Nuclear	WGD/segmental
	MD10G1275800	Chr10:36698263.36699996	801	266	29.5	5.10	Nuclear	WGD/segmental
	MD09G1235100	Chr09:29539556.29542546	753	250	27.4	5.47	Nuclear	WGD/segmental
	MD05G1265200	Chr05:40011060.40015621	1479	492	53.8	5.85	Nuclear	WGD/segmental
	MD10G1243000	Chr10:33776504.33781157	1482	493	53.2	6.05	Nuclear	WGD/segmental
	MD06G1091200	Chr06:21994255.21995556	888	295	33.6	5.49	Nuclear	WGD/segmental
	MD14G1112200	Chr14:18037604.18039324	948	315	35.6	5.05	Nuclear	WGD/segmental
	MD07G1131400	Chr07:18815423.18816949	1047	348	37.7	6.75	Nuclear	WGD/segmental
	MD01G1071300	Chr01:17602924.17604669	1050	349	37.7	8.16	Nuclear	WGD/segmental
	MD07G1280300	Chr07:34424687.34425782	837	278	31.1	7.10	Nuclear	WGD/segmental
	MD01G1210200	Chr01:30413680.30415409	1413	470	51.8	5.08	Nuclear	WGD/segmental
III	MD01G1168500	Chr01:27285229.27286785	687	228	25.7	7.72	Cytoplasmic/ Nuclear	WGD/segmental
	MD01G1215300	Chr01:30851473.30853003	1044	347	38.7	5.93	Nuclear	WGD/segmental
	MD07G1234600	Chr07:30813251.30815403	1029	342	37.4	5.48	Nuclear	WGD/segmental
	MD06G1104100	Chr06:24218568.24220572	1056	351	39.4	5.72	Nuclear	WGD/segmental
	MD07G1285200	Chr07:34745168.34746713	1029	342	38.0	5.50	Nuclear	WGD/segmental
	MD07G1285400	Chr07:34761880.34763425	1029	342	38.0	5.50	Nuclear	proximal
	MD07G1261100	Chr07:32679484.32683843	2988	995	113.2	8.64	Nuclear	dispersed
	MD14G1123000	Chr14:19736610.19738943	1068	355	39.9	5.66	Nuclear	WGD/segmental
	MD01G1078000	Chr01:18445739.18447924	1062	353	39.2	5.30	Nuclear	WGD/segmental
	MD07G1146900	Chr07:21450190.21453542	1071	356	39.7	5.48	Nuclear	WGD/segmental
	MD01G1168600	Chr01:27287662.27291066	912	303	34.2	5.71	Nuclear	WGD/segmental
	MD12G1189200	Chr12:27086797.27089009	1020	339	37.7	6.41	Nuclear	WGD/segmental
	MD04G1175600	Chr04:26669452.26671247	1035	344	38.2	6.01	Nuclear	WGD/segmental
	MD04G1175500	Chr04:26653551.26655300	1002	333	37.2	5.38	Nuclear	WGD/segmental
	MD12G1189900	Chr12:27224756.27226237	714	237	27.3	8.26	Extracellular	tandem
	MD07G1234700	Chr07:30816225.30818950	909	302	34.0	6.38	Nuclear	WGD/segmental
	MD12G1189700	Chr12:27213135.27215427	999	332	37.2	5.77	Nuclear	tandem
	MD12G1189600	Chr12:27190989.27194447	1110	369	40.6	5.69	Nuclear	tandem

## Data Availability

Not applicable.

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
