# Peer review of "Genome-Wide Analysis of the WRKY Gene Family in Malus domestica and the Role of MdWRKY70L in Response to Drought and Salt Stresses"

_genes, 2022, doi:10.3390/genes13061068_

Round 1

Reviewer 2 Report

The manuscript presented describes genome-wide analysis of the WRKY gene family in Malus domestica. Authors have done a great job on identifying, mapping, and analysis of apple WRKY genes. Also detailed study of the role of one of identified genes – Mdwrky70l – in response to drought and salt stress presents a great interest for creation of new varieties of cultivated plants resistant to adverse conditions.

Author Response

Dear Reviewer, thank you very much for taking the time to review this manuscript. We are very glad that you feel satisfy with our manuscript and accept it for publication. Thank you again.

Reviewer 3 Report

Manuscript ID genes-1760034

Qin et al., submitted the manuscript entitled “Genome-wide analysis of the WRKY gene family in Malus domestica and the role of MdWRKY70L in response to drought and salt stress“for publication consideration in Genes

This study provided a study of identifying 113 members of the WRKY genes from the apple genome GDDH13 and performed a comprehensive analysis, including chromosome mapping, phylogenetic, motif and collinearity analysis under drought and salt stresses. They used available Arabidopsis data sets to show the synteny relationship existed between AtWRKYs and MdWRKY. The manuscript provided valuable information with apple MdWRKYs function at whole genome level while exposed to drought and salt conditions especially MdWRKY70. 

The scientific soundness of this manuscript is acceptable except the experiment replications issue. It meets the aims and scope of Genes journal. 

This study is interesting; however several concerns should be considered.  

1. Focus and major contribution of this report is function of MdWRKY70, it should be highlighted and stated clearly.  Line 70,71,72 “In addition, transgenic tobacco plants were generated to analyze the biological role of one of the members, MdWRKY70L, under drought and salt stresses. Our results will be beneficial for revealing the role of WRKY proteins in apple responding to stressed conditions.” 

2. The prose and syntax of manuscript are required some attentions. It would be worthy to go through English editing.  

Line 13. 14: “WRKY transcription factor is a large regulatory protein family, which is unique in higher plants and is involved in the regulation of the plant stress response. In this study, we identified 113”. This sentence need to be improved and served as example of writing gap. 

3. “National Key Research and Development Program of China” and other grant agencies provided funding support for this research, not  as stated “Yangjun Zou, Fengwang Ma and Xiaoqing Gong provided financial support.” in line 426. 

4. In section of “Material and Methods-2.2. Plant Materials and Stress Treatments”,  one can’t find any biological repeats and stress treatments replications. Authors should address this in revision.   

Author Response

Response 1: Thank you for your suggestion. We have modified the description according to your comment in the revised manuscript in line 71-74 as follows:

Line 71-74: In addition, transgenic tobacco plants were generated to analyze the biological role of MdWRKY70L, which was induced by different stresses, under drought and salt stresses. Our results will be beneficial for revealing the role of MdWRKY70L proteins in apple responding to stressed conditions.

Response 2: Thank you for your suggestion. We are sorry that the writing was a bother for you, we have modified our manuscript with professional editing service and thoroughly revised our manuscript. All changes made in the manuscript are highlighted in yellow in the revised manuscript so that you can easily identify them. And we have changed the description in line 13-15 in the revised manuscript as follows:

Line 13-15: The WRKY transcription factors are kind of unique regulatory proteins in plants, which are important in the stress responses of plants. In this study, 113 WRKY genes were identified from the apple genome GDDH13 and a comprehensive analysis was performed

Response 3: Thank you for your comments. We have modified the author contribution in the revised manuscript in line 430-431 as follows:

Line 430-431: “Yangjun Zou, Fengwang Ma and Xiaoqing Gong provided funding acquisition.”

Response 4: Thank you for your suggestion. We have added the biological repeats and replications to the section 2.2 and 2.5 according to your comment in the revised manuscript in line 110-111 and line 145-147 as follows:

Line 110-111: All the treatments were performed with three biological replicates separately, with five plants in each replicates.

Line 145-147: The stress treatments were performed with at least three biological replicates with identical plant numbers in each replicates.